# Correlation of Neutralizing Antibodies (NAbs) between Sows and Piglets and Evaluation of Protectability Associated with Maternally Derived NAbs in Pigs against Circulating Porcine Reproductive and Respiratory Syndrome Virus (PRRSV) under Field Conditions

**DOI:** 10.3390/vaccines9050414

**Published:** 2021-04-21

**Authors:** Fu-Chun Hsueh, Sheng-Yuan Wang, Wei-Hao Lin, Chuen-Fu Lin, Chen-Yu Tsai, Chin-Wen Huang, Ning Sun, Ming-Tang Chiou, Chao-Nan Lin

**Affiliations:** 1Animal Disease Diagnostic Center, College of Veterinary Medicine, National Pingtung University of Science and Technology, Pingtung 91201, Taiwan; jim820723@gmail.com (F.-C.H.); april89000@gmail.com (S.-Y.W.); whlin@g4e.npust.edu.tw (W.-H.L.); cflin2283@mail.npust.edu.tw (C.-F.L.); a0911701551@gmail.com (C.-Y.T.); 2Department of Veterinary Medicine, College of Veterinary Medicine, National Pingtung University of Science and Technology, Pingtung 91201, Taiwan; a0986814635@gmail.com (C.-W.H.); putindog88@gmail.com (N.S.)

**Keywords:** maternal neutralizing antibody, correlates of protections, porcine reproductive and respiratory syndrome virus, viral load, field experiment, survival rate

## Abstract

Porcine reproductive and respiratory syndrome (PRRS), which is caused by a highly transmissible pathogen called porcine reproductive and respiratory syndrome virus (PRRSV), has caused severe problems, including reproductive disorders in sows and respiratory symptoms in nursery pigs worldwide, since the early 1990s. However, currently available PRRSV vaccines do not supply complete immunity to confront the viral infection. Elicitation of PRRSV-specific neutralizing antibodies (NAbs) during the preinfectious period has been deemed to be a feasible strategy to modulate this virus, especially in farms where nursery pigs are seized with PRRSVs. A total of 180 piglets in a farrow-to-finish farm that had a natural outbreak of PRRS were distributed into three groups based on the different PRRSV NAbs levels in their dams. In the present study, piglets that received superior maternal-transferred NAbs showed delayed and relatively slight viral loads in serum and, on the whole, higher survival rates against wild PRRSV infections. A positive correlation of maternal NAbs between sows and their piglets was identified; moreover, high NAbs titers in piglets can last for at least 4 weeks. These results provide updated information to develop an appropriate immune strategy for breeding and for future PRRSV control under field conditions.

## 1. Introduction

Since early 1990, epizootic porcine reproductive and respiratory syndrome (PRRS), which is caused by porcine reproductive and respiratory syndrome virus (PRRSV) strains, devastating the swine industry, has resulted in great economic losses in Europe, North America [1,2] and Asia [3,4,5]. PRRSVs, which belong to the family Arteriviridae and consist of a single-stranded, positive-sense RNA genome that is approximately 15 kilobases (kbs) in size, can be classified into two species, Betaarterivirus suid 1 (PRRSV 1, formerly called European type or type 1 PRRSV) and Betaarterivirus suid 2 (PRRSV 2, formerly called North American type or type 2 PRRSV) [6]. In Taiwan, PRRSV 2 has dominated the epidemic for over 20 years, although PRRSV 1 has also been recently discovered in a single farm [7,8]. The clinical signs, characterized by reproductive failure in sows and respiratory distress in nursery pigs, between these two species of PRRSV are analogous [8,9].

During the acute infection stage, PRRSV mainly replicates in alveolar macrophages and further expeditiously invades the lymphatic and hematological systems [10], leading to viremia that lasts for nearly a month and a series of antibody responses [11]. Viremia can be detected within 1 week after pigs are exposed to surroundings with PRRSVs [11]. Nevertheless, it takes nearly one month for PRRSV-infected pigs to elicit virus-specific neutralizing antibodies (NAbs) [12]. According to more detailed explorations of various antibodies, NAbs have been regarded as pivotal against PRRSV [13,14,15]. High NAbs titers tend to deliver passive protection and prevent transplacental PRRSV infection between sows and their offspring [13,15,16], which is consistent with the ideal management strategy in which sows with high-titer PRRSV-specific NAbs are conducive for piglets to receive competent immune protection via colostrum at an early stage against later exposure to the virus and its viremia [15,16]. Although vaccines were once considered to regulate PRRSVs, ensuing challenges incorporate feeble cross-protection against heterogeneous wild-type PRRSV strains [13]. Currently, the available modified live virus (MLV) vaccines have potential limitations and concerns regarding providing full protection, such as safety and efficacy, reversion of virulence, and recombination with wild-type PRRSV strains and MLV [17,18]. Thus, selection of sows with superior NAbs could be a promising avenue for long-term disease surveillance, especially on farms where PRRSVs circulate.

Neutralizing antibodies have been used under experimental conditions to delineate anti-PRRSV capabilities [15,16,19], whilst practical applications in pig farms are still lacking. To evaluate the immune responses of pigs against cycling PRRSV in the field, we divided piglets into three categories, including high, middle and low NAbs groups. Serum neutralizing antibodies and PRRSV viral loads were monitored every 2 weeks from 2 to 8 weeks of age. All dead pigs were used to calculate survival rates and to clarify the etiology. Our present results provide renewed understanding to develop a useful strategy for breeding and future PRRSV control under field conditions.

## 2. Materials and Methods

### 2.1. Experimental Program for Sows and Piglets

A clinical field trial was conducted in a 300-sow farrow-to-finish, 3-weekly batch system herd located in southern Taiwan. A total of 36 Landrace × Yorkshire sows from three batches, inoculated once every three months with PRRSV MLV vaccine (Ingelvac PRRSV MLV, Boehringer Ingelheim, St. Joseph, MO, USA), were exsanguinated at 3 weeks pre-parturition and were further separated into three different groups: the H (*n* = 15), M (*n* = 12) and L (*n* = 9) NAbs groups. The assorted criteria of NAbs were H (≥7 log_2_), M (6 log_2_ and 5 log_2_) and L (≤4 log_2_) groups, which were followed by previous experience with modifications [11,12,15,19]. After 36 sows farrowed, the offspring at 2 weeks of age that were first confirmed to be negative for PRRSV viremia were randomly selected by choosing five from each sow. The sum total of 180 2-week-old Landrace × Yorkshire × Duroc crossed-breed piglets were stochastically separated into three groups: high (*n* = 75), middle (*n* = 60) and low (*n* = 45) NAbs groups. No piglets were inoculated with PRRSV vaccines. The study was carried out in three series of batches, and each batch contained 60 piglets from three groups: high (*n* = 25), middle (*n* = 20) and low (*n* = 15) NAbs groups (Figure 1). No PRRSV challenges were performed in this study, but we assigned areas that were under the equivalent conditions in which an outbreak of PRRS had recently taken place to model natural viral infections in the field. Serum samples from piglets were collected in Vacutainer^®^ Plus Plastic SST^TM^ tubes with polymer gel (BD Medical, East Rutherford, NJ, USA) at 2, 4, 6, and 8 weeks of age to measure both PRRSV-neutralizing antibodies and viral loads.

### 2.2. Viral Neutralization Assay

PRRSV strain 763 (GenBank accession no. KY073240) was propagated in MARC145 cells (American Type Culture Collection, ATCC No. CRL-12231). Serum samples were first incubated at 56 °C for 30 min to inactivate the complement and were then diluted by 2^1^-fold to 2^10^-fold with fresh culture medium containing Dulbecco’s Modified Eagle Medium (Gibco, Gaithersburg, MD, USA), which was supplemented with 10% fetal bovine serum (FBS) (Gibco) and 1% penicillin-streptomycin solution (Biological Industries, Beit HaEmek, Israel). A mixture of PRRSV strain 763 (4000 TCID50/mL) and diluted serum samples of equal volumes were added to 96-well microplates and kept at 37 °C for 1 h; afterwards, trypsin-digested MARC145 cells (5 × 10^3^/well) were applied to the same microplates for 3 days of incubation. Cytopathic effects (CPE) were identified by immunofluorescence assay (IFA) as described below. The neutralizing titers were determined as the highest dilutions without CPE.

### 2.3. Immunofluorescence Assay (IFA)

After 3 days of incubation, the supernatants in microplates were first removed. The MARC145 cells were fixed by the 50% acetone (diluted by methonal) at 4 °C for 30 min, and air-dried for 2 h. Fixed cells were incubated with PRRSV monoclonal anti-mouse IgG1 (SDW17-F) (diluted 1:500 in 5% FBS) (RTI, LLC, Brookings, SD, USA) as primary antibodies at 37 °C for 1 h. After three washes with phosphate-buffered saline (PBS), fluorescein goat anti-mouse IgG (H+L) (2 mg/mL, diluted 1:1000 in PBS) (Invitrogen, Thermo Fisher Scientific, Waltham, MA, USA) was added at 37 °C for another 1 h to provide secondary antibodies. PRRSV-specific green, fluorescent signals were identified under an inverted fluorescence phase contrast microscope (Carl Zeiss AG, Oberkochen, Germany).

### 2.4. Quantification of PRRSV Viral Loads

Serum samples were examined by PRRSV M-gene-based quantitative reverse transcription polymerase chain reaction (RT-qPCR) with positive standards included to quantify viral titers [20]. Briefly, total nucleic acid extraction and complementary DNA (cDNA) synthesis adopting the MagNA Pure LC total nucleic acid isolation kit (Roche Diagnostics, Mannheim, Germany) and PrimeScriptTM RT reagent kit (Takara Bio Inc., Kusatsu, Shiga, Japan), respectively, were implemented following the manufacturers’ instructions. ZNA probe-based RT-qPCR was conducted using the primers (forward primer 5′-CATTCTGGCCCCTGCCCA-3′, reverse primer 5′-ACCACTCCYYGYTTDACAGCT-3′) and probes (NA probe 5′-FAM-CTCGTGTTGGGTGGCAGA-ZNA-4-BHQ1-3′, EU probe 5′-HEX-CGCTGTGAGAAAGCCCGG -ZNA-4-BHQ1-3′) on a LightCycler^®^ 96 Instrument (Roche Diagnostics) [8,20]. Thermal conditions with initial denaturation at 95 °C for 10 min with 45 cycles of 95 °C for 10 s, 55 °C for 10 s, and 72 °C for 15 s as the next step were used. The limit of detection for RT-qPCR was 0.96 log_10_ genomic equivalents (GE)/μL. The threshold for high PRRSV viral loads in serum samples was 4.2 log_10_ GE/μL for evaluating the presence of porcine respiration disease complex (PRDC) in asymptomatic and symptomatic PRRSV-infected pigs, as was reported earlier [20].

### 2.5. Survival and Etiological Analyses

All pigs were recorded from 2 to 10 weeks of age to calculate the survival rates among each group. Every dead pig underwent both histopathological examination and molecular diagnosis to figure out the cause of death, and further bacterial isolation was performed if necessary.

### 2.6. Statistics

Viral loads and NAbs titers were statistically calculated using SAS 9.4 (Statistical Analysis System, SAS Institute Inc., Cary, NC, USA) with one-way analysis of variance (ANOVA) in accordance with the assumption of additivity, tests of independence of error and tests of the homogeneity of variance as previously described [21]. Analysis of viral loads for evaluating the appearance of PRDC was cyphered by SAS 9.4 with two-way analysis of variance in conformity with the prerequisites of randomized compete block design (RCBD). Statistically significant differences were represented by *p*-values < 0.05 and/or *p*-values < 0.01. The correlations of NAbs titers among sows and their 2-week-old piglets were analyzed by linear regression, fitting the requirements of model adequacy checking. Significant differences were verified by log-rank test with GraphPad Prism 6 (GraphPad Software, San Diego, CA, USA).

## 3. Results

### 3.1. Evaluation of PRRSV NAbs among Varied Parities of Sows and Suckling Pigs

Conspicuous PRRSV fluorescent antigens were sprinkled within MARC145 cells during the viral neutralization assay (Appendix A) to accurately determine the titer of NAbs. The parities of 36 sows were separated into three categories: first parity, second to fourth parity, and over fourth parity. The neutralizing antibodies titers of each sample were as follows: 4.54 ± 1.65 log_2_, 6.67 ± 1.56 log_2_ and 6.00 ± 0.63 log_2_ (Figure 2). Significant differences in NAbs titers were observed among the first (4.54 ± 1.65 log_2_) and second (6.67 ± 1.56 log_2_) to fourth parities (*p* < 0.01). To evaluate the relevance of the NAbs titers between sows and their corresponding piglets, we utilized regression analysis (Figure 3). The NAbs titers in suckling piglets were positively affected by those in their birth mothers with a significantly positive correlation (r^2^ = 0.3377; *p* < 0.0001) (Figure 3).

### 3.2. Comparison of PRRSV NAbs in Pigs from Different Batches

The NAbs titers for each group at 2, 4, 6 and 8 weeks of age are shown in Figure 4. Compared with the low NAbs groups, the high NAbs groups manifested significant levels (*p* < 0.05) of eminent neutralizing antibodies at 2, 4 and 6 weeks of age in batches 1, 2 and 1+2+3 (Figure 4A,B,D). As anticipated, for pigs at 8 weeks of age, the titers of the high NAbs group in all batches except batch 1 plummeted to 1.69 ± 1.32 log_2_, 1.52 ± 1.53 log_2_, and 2.32 ± 2.01 log_2_ (Figure 4B–D). Statistically significant differences (*p* < 0.05) were annotated at 8 weeks of age in the high NAbs group (3.76 ± 2.21 log_2_ in batch 1, and 2.32 ± 2.00 log_2_ in batch 1+2+3) compared to the other two groups in batches 1 (0.75 ± 0.96 log_2_ in middle NAbs group and 0.15 ± 0.10 log_2_ in low NAbs group) and 1+2+3 (0.31 ± 0.64 log_2_ in middle NAbs group and 0.12 ± 0.11 log_2_ in low NAbs group) (Figure 4A,D). The middle NAbs groups, exclusively, showed significant differences (*p* < 0.05) at 2 weeks of age in batches 1 (5.55 ± 1.42 log_2_) and 1+2+3 (4.12 ± 1.37 log_2_) in comparison with the low NAbs groups (2.06 ± 1.30 log_2_ in batch 1, and 2.69 ± 1.57 log_2_ in batch 1+2+3) (Figure 4A,D), and at 4 weeks of age in batches 2 (1.90 ± 1.37 log_2_) and 1+2+3 (2.63 ± 1.53 log_2_) in comparison with the low NAbs groups (0.53 ± 0.24 log_2_ in batch 2, and 1.11 ± 0.64 log_2_ in batch 1+2+3) (Figure 4B,D). Clearly, the overall trend of NAbs titers uninterruptedly declined and fluctuated from 2 to 8 weeks of age.

### 3.3. Evaluation of PRRSV Viral Loads in Serum by Probe-Based RT-qPCR and the Presence of PRDC in Pigs

The titers of PRRSV viral loads in serum in each group at 2, 4, 6 and 8 weeks of age are depicted in Figure 5. The results for PRRSV 1 were all negative, as determined RT-qPCR. In batch 1, the mean viral load of the low NAbs group was 0.21 ± 0.30 log_10_ GE/μL at 4 weeks of age and shot up to 2.32 ± 1.66 log_10_ GE/μL at 6 weeks of age (Figure 5A). Compared to the high- and middle-NAbs groups in the same batch, the average viral loads were 0.24 ± 0.32 log_10_ and 0.42 ± 0.72 log_10_ GE/μL, respectively (Figure 5A). Statistically significant differences were noted at 6 weeks of age in batch 1 (2.32 ± 1.66, 0.42 ± 0.72, 0.24 ± 0.32 log_10_ GE/μL in the low, middle, high NAbs groups, respectively) (*p* < 0.05). In batches 2 and 3, the mean viral loads in all groups rose markedly at 6–8 weeks of age, but no significant differences were discovered in each group (Figure 5B,C). On the whole, the mean titers of PRRSV viral loads in the low NAbs groups were 1.70 ± 1.70 log_10_ GE/μL at 6 weeks of age and climbed dramatically at 8 weeks of age with a viral titer of 3.41 ± 0.77 log_10_ GE/μL (Figure 5D). In comparison with the high NAbs groups, the average viral load was 0.67 ± 0.85 log_10_ GE/μL at 6 weeks of age and rose up to 2.80 ± 0.99 log_10_ GE/μL at 8 weeks of age (Figure 5D). The numbers (and percentages) of pigs in each group (high, middle and low NAbs) whose serum viral loads exceeded 4.2 log_10_ GE/μL at 6 weeks of age were listed as follows: 3/73 (4.11%), 4/60 (6.67%) and 10/44 (22.73%), respectively (Figure 5E); those at 8 weeks of age were as below: 13/72 (18.06%), 10/57 (17.54%) and 15/44 (34.09%), respectively (Figure 5F). Statistically significant differences were calculated by two-way analysis of variance for the low NAbs groups (10/44, 22.73%) at 6 weeks of age as compared to the other two groups (3/73, 4.11% and 4/60, 6.67%) (*p* < 0.05).

### 3.4. Survival Rates and Diagnosis of Pathogens

To assess the protective efficacy of NAbs, we display the entire survival in Figure 6. Apart from batch 2, the high NAbs group exhibited favorable survival rates (≥90%) in all remaining batches compared to those (≤75%) in the low NAbs group, with significant differences (*p* < 0.05 or *p* < 0.01) (Figure 6A,C,D). Strikingly, no pigs in the high NAbs group in batch 3 died during the experiment. Although the survival rates for the three groups at 10 weeks of age in batch 2 were 80% (20/25), 75% (15/20), and 86.7% (13/15), no significant differences were observed (Figure 6B). Additionally, based on gross and histopathological examinations, 57.7% (15/26), 50% (13/26), and 30.8% (8/26) of all dead pigs were sequentially diagnosed with interstitial pneumonia, erosive to ulcerative colitis, and serositis, respectively. Those cases that were confirmed to have erosive to ulcerative colitis or serositis were subjected to bacterial isolation; consequently, 100% (13/13, all *Salmonella* spp.), and 25% (2/8, one *Glaesserella parasuis* and the other *Streptococcus suis*) of these were successfully isolated (data not shown). Interestingly, 8 out of 13 expired pigs that were diagnosed with salmonellosis belonged to the high and middle NAbs groups in batch 2. Regarding the molecular diagnoses, 80.8% (21/26) of all dead pigs had a PRRSV-positive result in serum samples based on RT-qPCR. The viral load titers in the high, middle and low NAbs groups were 3.17 ± 2.27 log_10_, 3.36 ± 2.08 log_10_ and 4.50 ± 1.45 log_10_ GE/μL, respectively, with no significant differences among groups.

## 4. Discussion

Elicitation of NAbs in pre-PRRSV infections has been believed to be a crucial strategy to eliminate the virus, shorten viremia, and further mitigate PRRSV-related syndromes [13,14,15,16,22]. However, the immune protection and efficacious duration of maternal antibodies with high NAbs titers from sows to piglets are still ambiguous. In this field study, we illustrated that piglets who received exceptional NAbs levels from sows had delayed and relatively inferior PRRSV viral loads in serum but showed superior survival rates at 10 weeks of age. The NAbs titers in those piglets evidently stayed at a comparatively high level, continuing for at least 4 weeks. Collectively, suckling piglets possessing higher-level maternal neutralizing antibodies could elicit augmented immune protection to prevail over PRRSV during the nursery phase.

Severe PRRSV viremia, which is defined as a viral load greater than 4.2 log_10_ (genomes/μL of serum), has been determined to be linked to the presence of PRDC in both asymptomatic and symptomatic PRRSV-infected pigs [20]. Based on the results of PRRSV loads in serum, the viral titers of pigs in the low NAbs groups all exceeded those of pigs in the high NAbs group (Figure 4A–D), but none of them surpassed the threshold of 4.2 log_10_ (genomes/μL of serum). It is foreseeable that no statistic difference was observed in the decreases of PRRSV viremia among three groups probably because no wild-type PRRSV challenge was carried out in this study. The data of serum PRRSV viral loads were initially used to verify two requisites. One was whether PRRSV exactly circulated in this farm or not, and the other was if experimental pigs could be naturally infected by latently circulating PRRSV. Surprisingly, when data were individually reexamined, 10 (22.73%) and 15 (34.09%) out of 44 pigs in the low NAbs groups at 6 and 8 weeks of age expressed PRRSV loads above 4.2 log_10_ GE/μL in serum (Figure 5E,F), but no typical PRRSV-associated symptoms were noticed in most of these individuals at that time. Furthermore, the survival curve plummeted especially at the low NAbs groups after 8 weeks of age. These may presage that asymptomatic PRRSV-infected pigs with high viral loads, i.e., 4.2 log_10_ GE/μL, are potentially suitable candidates for PRDC. In terms of the gross and histopathological evaluation, the gross lesions such as mottled appearance and consolidation texture and the histopathological lesion like interstitial pneumonia with mononuclear inflammatory infiltrate (macrophages and/or lymphocytes) accumulating in expanded alveolar septa would be considered to be related to PRRSV infection. However, these lesions are not specific enough to determine if they are caused directly by PRRSV. Different from the experimental challenge study, the condition of the field study is usually intricate since the bacterial invasion or environmental unpredictability cannot be straightforwardly avoided and excluded. Hence, further investigations, including PCR for monitoring other accompanying bacterial and/or viral pathogens, and immunohistochemistry to substantiate the direct causal among lesions, are needed.

PRRSV-specific NAbs, which are usually produced in one month after infection, have been viewed as an auspicious tool to address this tricky virus [11,12,22]. During the development of effective immune methods to control PRRS, researchers have found that pigs with NAbs titers above 3.4 log_2_ can significantly suppress viral replication [19]. After the relationship between PRRSV and NAbs is gradually disclosed, interest soon shifts to the effects depended on a doseof NAbs. PRRSV viremia might be persistently confined in pigs provided with NAbs levels over 3 log_2_; paradoxically, viruses continued to regularly replicate in the peripheral lymphatic tissues [15]. Intact protective immunity could be achieved only if pigs were given a higher titer, 5 log_2_ of PRRSV-specific NAbs from sows [15]. These decisive conceptions are practiced in the performance of vaccines to induce virus-specific NAbs. The PRRSV-neutralizing antibodies of vaccinated piglets increased to 5.5 log_2_ post challenge; simultaneously, the duration of PRRSV viremia was shorten to an average of 1 week [22]. In piglets, administration of both vaccine and maternal NAbs has been proposed to strengthen the immune responses against virulent PRRSV challenges [23], yet the exact workable titer of NAbs is still uncertain [13,17,18]. Therefore, the variations of maternally transferred NAbs in piglets by excluding the interference of vaccines is capable of unveiling this vague effect. To our knowledge, this is the first in situ field study which shed light on the positive correlations of maternal neutralizing antibodies between sows and their piglets and which elucidates the protective efficacy of neutralizing antibodies in nursery pigs against latently cycling PRRSV. No challenge trials were performed due to the simulation of real field conditions in which PRRSV occurred. Inevitably, bacterial incursion, such as *Salmonella* spp. would introduce an unpredictable bias, such as for batch 2 in this study; nevertheless, the final data represented that piglets receiving high titers of NAbs from their sows had superior survival rates on the whole. Most of the dead pigs (80.8%, 21/26) that were presented for histopathological and molecular diagnosis were proven to be succumbed to PRRSV infection, which lends credence to our field experiment. Quantification of PRRSV viral loads in the serum of dead pigs by RT-qPCR indicated that the titers of the low NAbs groups were higher than those of any other group, which supports the concept that pigs with high NAbs may have higher resistance to viral infections.

While the high and middle NAbs groups could induce superior neutralizing antibodies with respect to the low NAbs group, only those pigs in the groups with high NAbs titers exhibited exceptional survival rates at 10 weeks of age. Meanwhile, antiviral, modulatory, humoral and cellular immunity coregulating the dynamics of cytokines, such as TNF-α, IFN-γ, IL-10, TGF-β, IL-12, IL-4 and IL-6 might also play critical roles during the inflammatory period against PRRSV [10,12,13,14,15,24,25,26,27,28,29]. In addition, the existence of single nucleotide polymorphisms (SNPs) in host genes, such as GBP-1, Mx-1 and CD163, has also been attested to alleviate the viremia during PRRSV infections [30,31]. Future research can delve into whether sows and/or pigs equipped with both high titers of neutralizing antibodies and specific SNPs in their host genes could possess more comprehensive immune protection to combat PRRSVs.

## 5. Conclusions

To date, PRRSV remains a major problem in Taiwan with no effective control methods in pigs, especially at 5–8 weeks of age [32]. In our study, we successfully demonstrated that those pigs that acquired superior maternal-transferred NAbs showed tardy and relatively lower serum PRRSV loads but exhibited, on the whole, higher survival rates against latently circulating PRRSV although the bacterial incursion might affect the partial results of survival rates. No piglets were inoculated with PRRSV vaccines to boost immune responses. Due to numerous unanticipated problems with the current PRRSV vaccines, our results can provide practical information to establish more adequate immune strategies to modulate the disease and to set up a routine selection of gilts and/or sows hinged on NAbs titers for balancing the economic losses caused by PRRSV.

## Figures and Tables

**Figure 1 vaccines-09-00414-f001:**
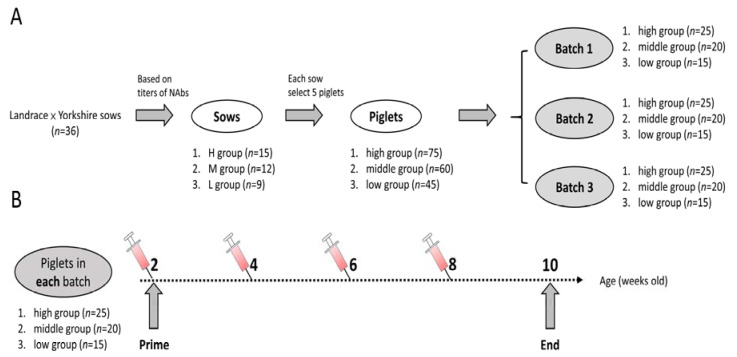
Schematic diagram of experimental arrangement (**A**) and protocol (**B**). Piglets (*n* = 180) in three different groups were randomly separated into three batches consisting of equal numbers (*n* = 60) and blooded at 2, 4, 6, 8 weeks of age.

**Figure 2 vaccines-09-00414-f002:**
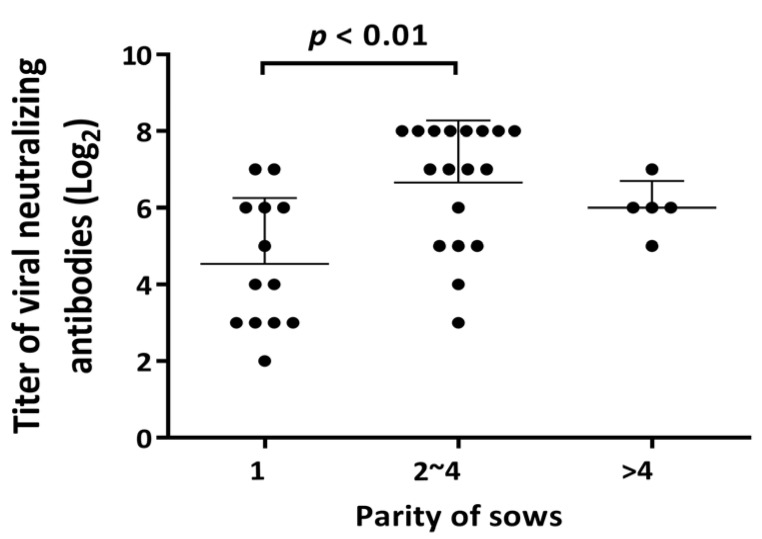
Detection of viral neutralizing antibodies in all sows. Data are displayed as the average NAbs titers, with error bars representing the standard deviations. The black dots represent each raw NAbs titer from the sows. Titers of neutralizing antibodies (NAbs) were as follows: 4.54 ± 1.65 log_2_ in first parity, 6.67 ± 1.56 log_2_ in second to fourth parities and 6.00 ± 0.63 log_2_ in over fourth parties. Significant differences in NAbs titers were observed between the first and second to fourth parities (*p* < 0.01).

**Figure 3 vaccines-09-00414-f003:**
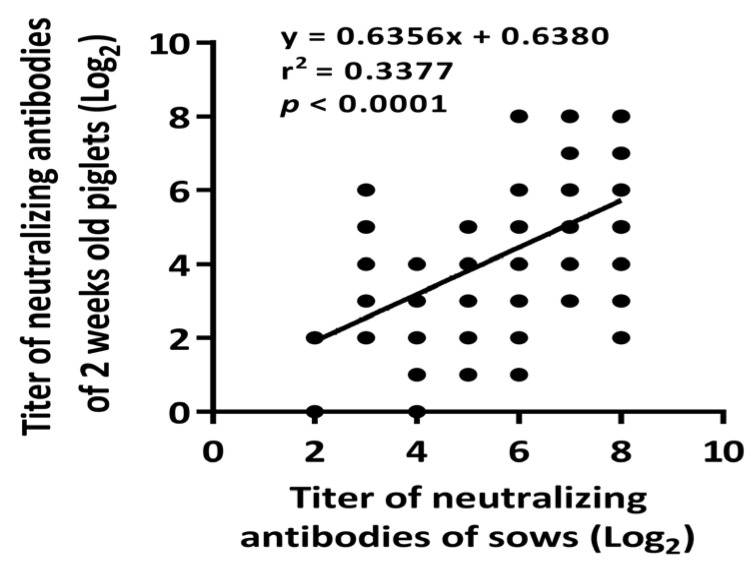
Correspondence of viral neutralizing antibodies between sows and their piglets. Black dots reflect each corresponding NAbs titer between sows and their piglets. The data correlation was analyzed by linear regression.

**Figure 4 vaccines-09-00414-f004:**
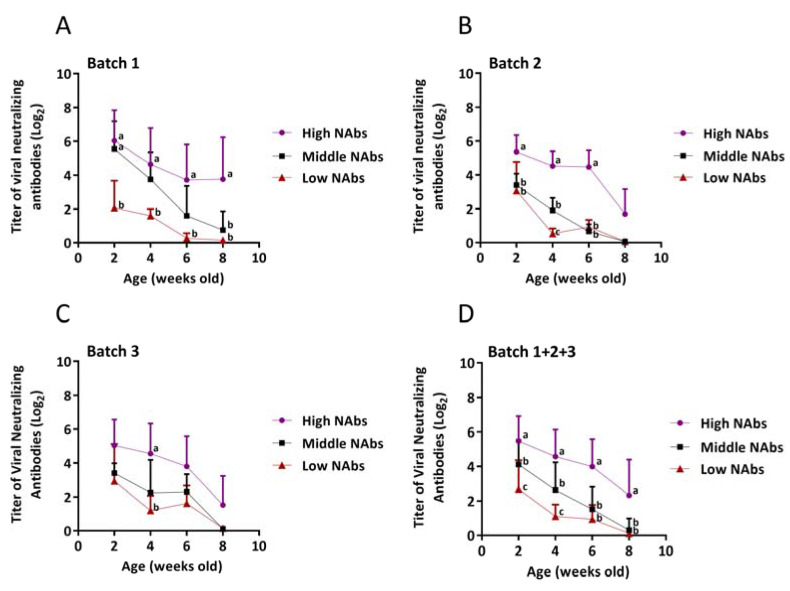
The mean titers of viral neutralizing antibodies (NAbs) in each group at 2, 4, 6, and 8 weeks of age (**A**–**D**). Error bars indicate standard deviations. (**A**) Titers of NAbs at each week of age between high and low NAbs were as listed below, 6.04 ± 1.61 log_2_ and 2.06 ± 1.30 log_2_, 4.64 ± 1.91 log_2_ and 1.60 ± 0.32 log_2_, 3.72 ± 1.86 log_2_ and 0.26 ± 0.25 log_2_, and 3.76 ± 2.21 log_2_ and 0.15 ± 0.10 log_2_. (**B**) NAbs titers at 2, 4, 6 weeks of age between high and low NAbs were as follows: 5.36 ± 0.89 log_2_ and 3.06 ± 1.38 log_2_, 4.52 ± 0.78 log_2_ and 0.53 ± 0.25 log_2_, and 4.47 ± 0.86 log_2_ and 0.93 ± 0.34 log_2_. (**C**) Titers of NAbs at four weeks of age between high and low NAbs were 4.56 ± 1.58 log_2_ and 1.20 ± 0.71 log_2_. (**D**) Viral NAbs titers at each week of age between high and low NAbs were reported, 5.48 ± 1.38 log_2_ and 2.69 ± 1.57 log_2_, 4.57 ± 1.50 log_2_ and 1.11 ± 0.64 log_2_, 4.00 ± 1.54 log_2_ and 0.94 ± 0.78 log_2_, and 2.32 ± 2.01 log_2_ and 0.12 ± 0.11 log_2._ Statistically significant differences were recorded as a, b, and c (*p* < 0.05).

**Figure 5 vaccines-09-00414-f005:**
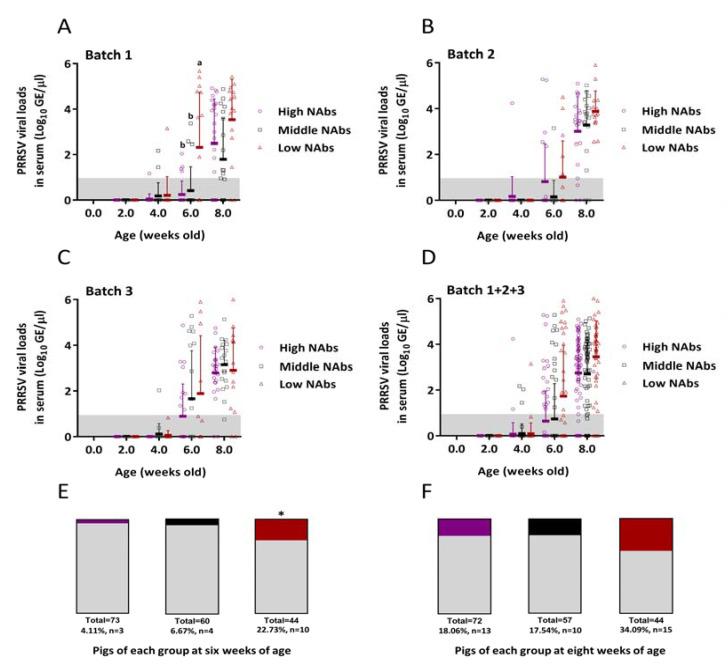
PRRSV viral loads in serum as detected by RT-qPCR (**A**–**D**). The mean values of PRRSV genomic equivalents (GE) were transformed into log_10_ GE/μL. The error bars indicate standard deviations. The limit of detection for RT-qPCR was 0.96 log_10_ GE/μL marked as the gray zone. PRRSV viral loads in the low, middle and high NAbs groups at six weeks of age in batch 1 were as follows, 2.32 ± 1.66, 0.42 ± 0.72 and 0.24 ± 0.32 log_10_ GE/μL, respectively. Statistically significant differences were present between a and b (*p* < 0.05). Numbers (and percentages) of pigs in the high (purple area), middle (black area) and low (red area) NAbs groups whose viral loads in serum exceeded 4.2 log_10_ GE/μL are shown in the stacked bar graphs (**E**,**F**). The gray areas of the bar graphs represent the remaining pigs whose serum viral loads were below 4.2 log_10_ GE/μL in each group. * indicates significant differences (*p* < 0.05).

**Figure 6 vaccines-09-00414-f006:**
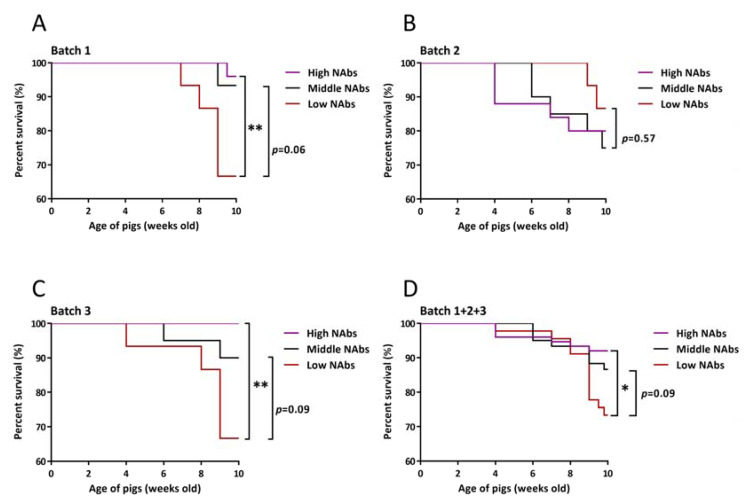
Survival curves of each group in different batches (**A**–**D**). Data for dead pigs were evaluated daily. The survival rates in the high and low NAbs groups were presented, 96% and 67% in batch 1, 80% and 87% in batch 2, 100% and 67% in batch 3, and 92% and 73% in batch 1+2+3, respectively. * and ** indicate significant differences, *p* < 0.05 and *p* < 0.01, respectively.

## Data Availability

The data presented in this study are available on request from the corresponding author.

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
