# Peer review of "Correlation of Neutralizing Antibodies (NAbs) between Sows and Piglets and Evaluation of Protectability Associated with Maternally Derived NAbs in Pigs against Circulating Porcine Reproductive and Respiratory Syndrome Virus (PRRSV) under Field Conditions"

_vaccines, 2021, doi:10.3390/vaccines9050414_

Round 1

Reviewer 1 Report

The study intriguingly suggested the potentially effective protection against PRRSV for piglets by Nab acquired from sow with high Nab titers. The study was well designed with solid techniques and analyses being conducted. However, the interpretation of results, conclusion and discussion are yet to be refined.

The use of words superior protection in the title and in drawing conclusions is exaggerating, misleading, and lacking of specificity. The data showed significantly higher Nab titer in piglets from sows with high level of Nabs, which is not surprising at all. However, the reduction of serum viral loads was far from being substantial; the survival rates were too variable among batches to support the high Nab sow strategy being practically viable.

The observations are valuable and will be more useful if the authors can interpret data in specifically accurate ways, make conclusions unbiasedly, and importantly discuss the utility of the phenomenon in the context of technical feasibility, cost and profitability. 

Author Response

Point by point response letter

Reviewer #1

1. The study intriguingly suggested the potentially effective protection against PRRSV for piglets by Nab acquired from sow with high Nab titers. The study was well designed with solid techniques and analyses being conducted. However, the interpretation of results, conclusion and discussion are yet to be refined.

Response: We thank the Reviewer #1 for the insightful point.

2. The use of words superior protection in the title and in drawing conclusions is exaggerating, misleading, and lacking of specificity.

Response: We thank the reviewer #1 for the insightful point. We have modified our previous title into “Correlation of neutralizing antibodies (NAbs) between sows and piglets and evaluation of protectability associated with maternally derived NAbs in pigs against circulating porcine reproductive and respiratory syndrome virus (PRRSV) under field conditions” to express our study accurately and specifically. We also add a proviso in our conclusion in the revision to clarify that some unpredictable bacterial incursion could not be excluded in this study. (Line 343-345)

3. The reduction of serum viral loads was far from being substantial; the survival rates were too variable among batches to support the high Nab sow strategy being practically viable.

Response: We appreciate this detail that Reviewer # 1 highlighted. The reason why the reduction of serum viral loads is not obvious among each group is probably because no PRRSV challenge is performed in this study. Our objective is to evaluate, exclusively, the protectability of maternal neutralizing antibodies against PRRSV and to practically apply this strategy in the pig farm. Therefore, the artificially PRRSV challenge would not be suitable especially in field studies. In addition, the data of PRRSV viral loads were initially used to verify two requisites. One was whether PRRSV exactly circulated in this farm or not, and the other was if all pigs in this study could be naturally infected by latently circulating PRRSV. However, when we carefully analyze this result, we find out that the serum viral loads in the high NAbs groups are tardy and relatively lower as compared to those in the low NAbs groups. We have added some description in Line 276-281 in the revision to illustrate this question. Next, the survival rates in the high NAbs groups are significantly higher than the low NAbs groups in batch 1 (Fig 5A), batch 3 (Fig 5B) and batch 1+2+3 (Fig 5D) (p < 0.05 and/or p < 0.01). We had explained the reason why the survival rates in the high NAbs groups of batch 2 are relatively low (Line 318-321). Even if we take the results of batch 2 into the calculation of comprehensive survival rate, namely results of batch 1+2+3, the final data still represents that piglets receiving high titers of NAbs from their sows had superior survival rates (Fig 5D). Thus, our selective strategy of high titers of neutralizing antibodies could still be a practically viable method for breeding and future PRRSV control under field conditions. (Line 343-349)

4. The observations are valuable and will be more useful if the authors can interpret data in specifically accurate ways, make conclusions unbiasedly, and importantly discuss the utility of the phenomenon in the context of technical feasibility, cost and profitability.

Response: We thank the reviewer #1 for these precise suggestions. The clarified information has been added in the abstract, introduction, discussion and conclusion in the revision. The aim of this study is to provide a new method to combat PRRSVs in the field since PRRSV remains a major problem in Taiwan with no effective control methods in nursery pigs. Our team often helps to calculate the potentially economic losses of the pig farm caused by PRRSVs and to evaluate if this pig farm is suitable for this selective strategy. To illustrate, if the owner can retain the sows equipped with high NAbs titers, and each sow could have sixth parity of piglets in general, the mean cost of this method is approximately $3.5 US dollar per parity per sow (about $21 US dollar per serum neutralization test against PRRSV in our diagnostic center). The farm owners can measure costs and profits in their own situation and make the final decision by themselves. We set up this selective strategy to help to balance the economic losses caused by PRRSV due to numerous unanticipated problems with the current PRRSV vaccines. The supplementary description had added to the Line 348-349 in the revision.

Reviewer 2 Report

Major comments:

The authors have shown, under field conditions, piglets with higher maternal Neutralizing antibodies, show better survival against wild PRRSV infections. Overall, it’s a well-designed study. However, there are a few comments that need further clarification/explanation

  • The title should explicitly mention the word correlation/correlates; this is important as this is a field study, and no challenge has been done. Also, the design of the study does not prove causation
  • For explaining method 2.1, I would suggest including a flow chart with number of participants (sows/piglets), their segregation into groups and the various batches. This would make it easier to follow
  • It was difficult to find whether the primers for real time PCR were directed against both PRSSV strains or not. Also after PCR amplification, did the authors sequence the PCR products?
  • Figure 4 – Line plots do not show the distribution of data. The authors should also show the time point viral titers in a scater plot with bar/column plot. This would show the distribution of viral titers in piglets. For each time point, there would be 3 pairs of bars/column with dots.
  • Can the authors also show pathological evaluation of piglets that died of PRRSV infection in various groups? If there are no more samples then it should be discussed in the discussion section
  • Limitations of the study should be discussed in more detail – what are the confounding factors in this study? In terms of viral titers, there isn’t much different between the groups – why is that? What are the limitations of field study vs challenge study in the context of NAbs for PRRSV? Limitations of using serum samples for viral load study vs lung/oral.

Minor comments:

The authors might want to cite articles to this study like - https://doi.org/10.3390/vaccines9020106

https://dx.doi.org/10.1371%2Fjournal.pone.0223060

Line 65: “have been applied” to “have been used/considered”

Line 70: reword the sentence – addressed is not the right word here

Line 109: the sentence is grammatically incorrect

Line 153: What to the authors want to convey by the usage of fluorescent antigens? The message is unclear here.

Line 178: How have the authors calculated mean and standard deviation? One of the values has standard deviation more than mean which would mean that the lowest value comes in negative and log values cannot be negative. And the authors should also mention in brackets whether its Mean ± standard error/standard deviation wherever necessary in results section

Line 292: Is susceptible the correct word here? Or did the pigs most likely succumb due to PRRSV infection?

Author Response

Point by point response letter

Reviewer #2

1. The authors have shown, under field conditions, piglets with higher maternal Neutralizing antibodies, show better survival against wild PRRSV infections. Overall, it’s a well-designed study. However, there are a few comments that need further clarification/explanation

Response: We thank the Reviewer #2 for the insightful point.

2. The title should explicitly mention the word correlation/correlates.

Response: We thank the reviewer #2 for the insightful point. We have modified our previous title into “Correlation of neutralizing antibodies (NAbs) between sows and piglets and evaluation of protectability associated with maternally derived NAbs in pigs against circulating porcine reproductive and respiratory syndrome virus (PRRSV) under field conditions” to express our study accurately and specifically.

3. For explaining method 2.1, I would suggest including a flow chart with number of participants (sows/piglets), their segregation into groups and the various batches. This would make it easier to follow.

Response: We apologize for the unclear description. To make Method 2.1 more clearly, we have added a new Figure 1 which is related to the experimental arrangement and protocols. (Line 96-99)

4. It was difficult to find whether the primers for real time PCR were directed against both PRRSV strains or not. Also after PCR amplification, did the authors sequence the PCR products?

Response: We thank the reviewer #2 for the insightful point. As to the primers of real-time PCR, we used the same primer pair but different probes to target both PRRSV 1 (European type) and PRRSV 2 (North American type). The results could be specifically distinguished by different fluorescent signals of ZNA probes. We designed the primer pair which is based on the M gene of PRRSVs. This method had been done in our previous research [8, 20](Line 135). After qPCR amplification, we had selected one positive sample from each group at each age and sent for sequencing to exclude any non-specific result. All sequences share the similar results, which are closely related to the PRRSV 2 strain.

5. Figure 4 – Line plots do not show the distribution of data. The authors should also show the time point viral titers in a scatter plot with bar/column plot. This would show the distribution of viral titers in piglets. For each time point, there would be 3 pairs of bars/column with dots.

Response: We thank the reviewer #2 for the insightful point, we have modified our Figure 5A-5D into the pattern of a scatter plot.

6. Can the authors also show pathological evaluation of piglets that died of PRRSV infection in various groups? If there are no more samples then it should be discussed in the discussion section. Limitations of the study should be discussed in more detail – what are the confounding factors in this study? In terms of viral titers, there isn’t much different between the groups – why is that? What are the limitations of field study vs challenge study in the context of NAbs for PRRSV? Limitations of using serum samples for viral load study vs lung/oral.

Response: We appreciate this detail that Reviewer # 2 highlighted. First, during the necropsy of pigs, we did the gross examination first. The gross lesion including mottled appearance and consolidation texture would be considered with potential PRRSV infection. Then, we did the histopathological examination to make sure if the lesions of histopathological examination could be correlated with those of gross examination. The histopathological lesions are characterized by interstitial pneumonia with mononuclear inflammatory infiltrate (macrophages and/or lymphocytes) accumulating in expanded alveolar septa. However, these PRRSV-related lesions are not specific enough to determine if these lesions are caused directly by PRRSV. It might have chances to be caused by bacterial or other viral infection, such as Mycoplasma spp. or PCV2 especially in the field study. We have supplemented detailed situation and limitations of the field study (Line 287-297). Thus, we finally decided to combine the results of both histopathological and molecular diagnosis to prove that the dead pigs were succumbed to PRRSV infection (Line 321-324). Second, the reason why the reduction of serum viral loads is not obvious among each group is probably because no PRRSV challenge is performed in this study. Our objective is to evaluate the protectability of maternal neutralizing antibodies exclusively against PRRSV and to practically apply this strategy in the pig farm. Therefore, the artificially PRRSV challenge would not be suitable especially in field studies. In addition, the data of PRRSV viral loads were initially used to verify two requisites. One was whether PRRSV exactly circulates in this farm or not, and the other was if all pigs in this study could be naturally infected by latently circulating PRRSV. However, when we carefully analyze this result, we find out that the serum viral loads in the high NAbs groups are tardy and relatively lower as compared to those in the low NAbs groups. We have updated some description to illustrate this question (Line 276-281). Third, the limitations between field study and challenge study are almost discussed above and also written in the revision. The artificially PRRSV challenge would not be suitable especially in field studies. Once no challenge is performed, the differences of viral titers would be inconspicuous to be differentiated between each group (Line 276-281 and 293-295). Finally, we utilize the serum sample instead of lung or oral fluid due to the following reasons. Previous research pointed out that PRRSV viremia could reach a high peak within approximately 3 days and maintain this peak for at least two weeks; however, the titer of viral loads in tissues would decrease apparently within two weeks [2-3]. Furthermore, the serum representing the systemic condition of the pig would be more precise than the lung expressing only the local respiratory condition. The quantification of viral loads is also another problem because viral loads in lung can only be quantified relatively as compared to the host genes, but the serum can be quantified absolutely, directly and more easily especially in the field study. If we have to unify the criteria of the quantification, the serum would be more appropriate than the lung. Also, those alive pigs cannot be able to measure the titers of viral loads by the lung. As to the oral fluid, this method is more humane and accords with the animal welfare without doubts. Unfortunately, the actual manipulation would not be feasible because the oral fluid is more likely to be used in monitoring groups of pigs. It would be tough for us to avoid the interference of other pigs in the field, which contains 60 pigs in each batch. More importantly, the amount of oral fluid is also not to be unified as compared to that of the serum. Thus, we finally choose the serum to measure the PRRSV viral loads.

7. Minor comments: The authors might want to cite articles to this study like - https://doi.org/10.3390/vaccines9020106https://dx.doi.org/10.1371%2Fjournal.pone.0223060

Response: We thank the reviewer #2’s suggestion, we have updated the references. (Line 333)

8. Line 65: “have been applied” to “have been used/considered”

Response: We have modified as suggested. (Line 65)

9. Line 70: reword the sentence – addressed is not the right word here

Response: We have modified as suggested. (Line 70)

10. Line 109: the sentence is grammatically incorrect.

Response: We have modified as suggested. (Line 115-117)

11. Line 153: What to the authors want to convey by the usage of fluorescent antigens? The message is unclear here.

Response: We apologize for the unclear description. We used the IFA to specifically identify the cytopathic effects (CPE) of PRRSV-infected MARC-145 cells (Line 115-117, 160-161). Truly, it is not really necessary to do this. We can directly judge by the morphology of infected cells. However, based on our previous experience, CPE sometimes would be confused with naturally dead cells. To pursue more accurate results, we finally use the IFA to help for identifying the CPE. (Line 161)

12. Line 178: How have the authors calculated mean and standard deviation? One of the values has standard deviation more than mean which would mean that the lowest value comes in negative and log values cannot be negative. And the authors should also mention in brackets whether its Mean ± standard error/standard deviation wherever necessary in results section

Response: We apologize for the unclear description. We import the individual value into the software, SAS 9.4 (Statistical Analysis System, SAS Institute Inc., Cary, NC, USA) to help for calculating both mean value and standard deviation. There is no direct relationship between mean value and standard deviation because the mean is the average of the sum of data, while the standard deviation is calculated by taking the square root of the average of the values of squared deviations. Statistically, there is no limit on standard deviation in association with mean value. And because this is a field study with PRRSV natural infection, the data generally shows in the normal distribution. This suggests that the distribution could be apparently distanced or differentiated and further increases the opportunity of implicitly elevating the standard deviation. It is true that log values cannot be negative; however, we might not really use the mean to minus the standard deviation. We agree with the idea that this could be a controversy, so the detection limit of RT-qPCR has been added in Line 137 and 231, which suggests that the mean value of viral loads lower than 0.96 log10 GE/μl could not be reliable. The representative graphic of neutralizing antibodies is similar to previous study by using the logarithmic transformation [4]. The reason why we prefer to use the line chart is to clearly strengthen the variation trend between each age among different groups, which coheres to our research aim, “evaluation of protectability associated with maternally derived NAbs in pigs.” Thanks for reminding us to mention in brackets with its mean ± standard deviation. We have checked all results parts and added them in the revision.

13. Line 292: Is susceptible the correct word here? Or did the pigs most likely succumb due to PRRSV infection?

Response: Many thank you for this correction. We have corrected this word in text. (Line 323)

Round 2

Reviewer 1 Report

I am satisfied with the revision and wish the authors the great success in further development of the technology.

Author Response

We thank you for your patient and thorough review of our manuscript, which have increased the quality of the final manuscript. 

Reviewer 2 Report

I would like to thank the authors for addressing my comments.

Author Response

(The authors gave the same response as above.)
